# Cell Cycle Dynamics in the Microalga *Tisochrysis lutea*: Influence of Light Duration and Drugs

**DOI:** 10.3390/cells13221925

**Published:** 2024-11-20

**Authors:** Laura Pageault, Aurélie Charrier, Bruno Saint-Jean, Gaël Bougaran, Francis Mairet, Sabine Stachowski-Haberkorn

**Affiliations:** 1Laboratoire GenAlg Nantes, Unité Physiologie et Toxines des Microalgues Toxine (PHYTOX), IFREMER F-44311 Nantes, France; bruno.saintjean@ifremer.fr; 2Laboratoire PhysAlg Nantes, Unité Physiologie et Toxines des Microalgues Toxine (PHYTOX), IFREMER F-44311 Nantes, Francegael.bougaran@ifremer.fr (G.B.); francis.mairet@ifremer.fr (F.M.); sabine.stachowski.haberkorn@ifremer.fr (S.S.-H.)

**Keywords:** synchronization, photoperiod, nocodazole, aphidicolin, hydroxyurea, sizer, timer, commitment point

## Abstract

Our investigation into *Tisochrysis lutea*’s cell cycle regulation involved natural and chemical synchronization methods to maximize their proportion at the division phase (G_2/M_). Hence, cultures were grown under different light/dark cycles (24:0, 12:12, and 8:16 h) to assess the impact of extended dark periods on cell division. Flow cytometry analyses of the cell cycle revealed that extending the dark phase resulted in a higher number of cells entering G_2/M_. However, this remained a minority within the overall culture (peaking at 19.36% ± 0.17 under an 8:16 h L/D cycle). To further enhance synchronization, chemical agents (nocodazole, hydroxyurea, and aphidicolin) were tested for their efficacy in blocking specific cell cycle stages. Only aphidicolin successfully induced significant G_2/M_ accumulation (>90%). The commitment point for cell division was examined by exposing cultures to varying light durations (0 to 8 h) and measuring cell concentration and size distribution every 4 h. Our findings identified a critical minimum cell size (“sizer”) of approximately 56.2 ± 0.6 µm^3^ and a required minimal light exposure (“timer”) of 4 h to reliably trigger cell division. These findings highlight key conditions needed for optimal division of *Tisochrysis lutea*, offering more controlled and efficient cultivation strategies for future biotechnological applications.

## 1. Introduction

Phytoplankton represent true ecosystem engineers that provide numerous supporting services fundamental to regulating and sustaining life on our planet. Indeed, phytoplankton significantly influence matter dynamics, including primary production, biogeochemical cycles, and nutrient distribution, not only in aquatic but also in terrestrial ecosystems [1]. Their contribution to nearly half of the Earth’s oxygen production [2] and regulating role in the global climate highlight their immense impact on the planet’s ecological balance.

The extensive distribution and diversity of this group of organisms, coupled with their thriving presence within oceanic ecosystems, establish them as essential biological drivers of our biosphere.

Diatoms, dinoflagellates, and haptophytes stand out as the most dominant groups driving seasonal blooms within marine phytoplankton [3,4]. In addition to serving as a fundamental physiology model, these groups contain species harboring unique characteristics such as high-value carotenoids (i.e., fucoxanthin, astaxanthin) for human health [5,6] or large amounts of proteins and fatty acids useful for animal and aquatic feed [7,8] with high biotechnological promise.

Within haptophytes reside a distinctive species, *Tisochrysis lutea*, a member of the Isochrysidaceae family known for its non-calcifying aspect. Easy to grow in laboratory conditions, *T. lutea* has been historically used in aquaculture for its abundance of polyunsaturated fatty acids, particularly docosahexaenoic acid (DHA) [9,10]. *T. lutea* is now a focal point in biotechnology research. Notably, it boasts substantial levels of fucoxanthin, one of the most economically appealing carotenoid pigments, recognized for its various properties, including antioxidative, anti-inflammatory, anticancer, and antimicrobial effects, to mention a few [6,11,12]. Due to these biotechnological prospects, its rapid adeptness, and efficient growth across diverse conditions [13,14], considerable knowledge has been amassed for this species. Nevertheless, its life cycle remains to be described.

The life cycle of a microalga is intricately tied to its cell cycle due to its unicellular nature, involving growth, division, and potentially the formation of spores or gametes [15].

Understanding the life cycle of microalgae is vital for grasping their growth dynamics, reproductive strategies, etc. It is essential for optimizing their biotechnological applications, ranging from aquaculture and biofuel production to pharmaceutical development. The cell cycle, a ubiquitous and intricate process, orchestrates cell growth, proliferation, and cell type differentiation and regulates events necessary for direct reproduction [16,17]. Two sequential phases constitute this cycle: interphase (G_1_, S, G_2_ phases), marked by cellular growth, and mitosis (M phase). To uphold the cell’s integrity throughout the pivotal phases of the cell cycle, a regulation system with checkpoints exists at transitions between phases. These checkpoints encompass a series of dependencies or prerequisites that the cell must fulfill to proceed through the cycle, such as growth, precise chromosome segregation during mitosis, vigilance over DNA replication, or cell size [18]. Among the most well-studied are the cell size checkpoints. Studies on mammalian cells [19] or amphibian oocytes [20] revealed a dependency of a critical cell size between the S phase and mitosis; this size control point is called the “commitment point” (CP) and is considered the “start” of cell division [21]. Two main controllers have been proposed to constitute this regulatory step: a “sizer”, which assesses whether the cell has attained the critical size for progression, and a “timer,” which gauges if the cell has spent a sufficient duration in the previous growth phase [22,23]. Thus, the rate at which a cell attains commitment is tightly correlated with its growth rate, itself influenced by inherent parameters (i.e., physiological status) combined with environmental factors [24]. However, different species and cell types vary widely in the intervention of these checkpoints within the cell cycle.

Although the genes responsible for these regulatory processes are highly conserved across eukaryotes and yeast, the regulation of the cycle exhibits notable differences in planktonic organisms [15,25].

In autotrophic microalgae, it has been proven that growing phases (G_1_, G_2_) are predominantly regulated by the rate of photosynthesis, with light serving as a crucial energy source [26,27]. The daily fluctuations in external light conditions (i.e., day and night), coupled with the internal circadian clock [28], play a pivotal role in orchestrating the life cycle events of microalgae, including the cell cycle, growth, and reproduction. This dependence proves beneficial for coordinating light-dependent processes like photosynthetic growth during the day and light-independent mitotic division at night [29].

Consequently, many species of microalgae cultivated under a light/dark cycle exhibit synchronized patterns of growth [30]. Synchronization, in this context, refers to methods that induce all cells to advance synchronously and in a coordinated manner through the distinct phases of the cell cycle [31]. The cell synchrony of cultures proves invaluable for accurately investigating cell progression through each phase of the cell cycle and understanding the underlying molecular mechanism of cellular processes [32,33,34,35].

Light, specifically the photoperiod, has been identified by Heldt et al., 2020 [34], and by Lemaire et al., 1999 [36], as a pivotal factor influencing the size control mechanism at CP that initiates cell division in *Chlamydomonas* cells, responding to various light/dark patterns. While natural synchrony is observed in certain microalgal cultures, induction of cell synchrony can also be achieved by limiting light (intensity and/or duration), as the deprivation will entail cell cycle arrest at the G1 phase [37,38].

As demonstrated by Yee and Bartholomew (1988) [39], *Euglena gracilis* cultures need at least 6 h of daylight exposure to progress beyond the commitment point (G1-S) in their cycle; otherwise, the cycle stops. If the light factor is not a determining factor, the addition of chemical agents arresting the cell cycle at a specific stage also provides effective and reversible culture synchronization [40,41].

Despite the proven effectiveness of these methods, the description of cell cycles in microalgae has been limited since the early 2000s, with only one or two species (e.g., *Chlamydomonas reinhardtii* or *Scenedesmus* spp.) serving as model organisms [21,23,24,42,43,44,45]. Currently, numerous studies focus on easily manipulable biological models, known for their ease of cultivation and rapid growth, to improve these models for industrial purposes. However, despite the insights gained from these investigations, fundamental gaps in understanding the biology of the targeted species remain. Exploring essential cellular mechanisms such as the cell cycle, proliferation, and differentiation opens up untapped potential for species improvement through genetic crosses and hybrid strains. While cell cycle studies in model microalgae like *T. lutea* are still emerging, this research addresses key gaps by focusing on the factors that influence division timing and phase synchronization, areas previously underexplored in the context of microalgal biotechnology. This holds for numerous species, including our biological model, *T. lutea*.

The biotechnology industry is increasingly leaning towards the utilization of hybrid strains tailored to changing industry and research needs [46,47,48]. These genetically cross-bred varieties combine desired traits from different strains, each lacking these characteristics individually. Crossbreeding fosters and preserves genetic diversity, countering the reduction caused by genetic selection or mutagenesis techniques [49].

To achieve this kind of genetic crossbreeding with *T. lutea* strains, it is essential to understand and master the reproductive events occurring during its cell cycle.

Hence, this study endeavors to delineate the cell cycle of this haptophyte for the first time.

Firstly, we aimed to investigate *T. lutea*’s cellular cycle by comparing natural and chemical cell synchronization, using light and various drugs, respectively. Secondly, we looked at the effect of increasing illumination time on cell cycle progression. Inspired by the growth-interruption experiments run by Heldt et al. (2020) [34], we aimed to highlight the *T. lutea* size control at the CP.

## 2. Material and Methods

### 2.1. Microalga Strain

The marine microalga *Tisochrysis lutea* strain 927.14 was provided by the Culture Centre of Algae and Protozoa (CCAP, Scotland, UK). The absence of bacteria in the cultures was checked with epifluorescent microscopy (Olympus BH2-RFCA, Tokyo, Japan) after staining with SYBR Green I (Invitrogen™ SYBR™ Green I Nucleic Acid Gel Stain, Reference: S7585, ThermoFisher Scientific, Waltham, MA, USA).

### 2.2. Culture Conditions

*T. lutea* cultures were maintained in sterile Conway medium [50] at 21 ± 1 °C, in a thermostatic chamber (MLR-352-PE, PHCBI, Tokyo, Japan) at 140 µmol m^−2^ s^−1^ (Quantometer Li-Cor Li-250 equipped with a spherical sensor), with a light: dark cycle (L:D) of 8:16 h. Cultures were grown in 250 mL Erlenmeyer sterile glass flasks previously autoclaved for 20 min at 121 °C and then filled with 150 mL of sterile culture medium. Cultures were diluted weekly at 1/50 dilution to maintain an exponential growth phase and to have cultures ready to use for experiment inoculation.

### 2.3. Photoperiod Experiment

A first set of independent experiments was carried out to compare the effect of continuous illumination (24:0 L:D) versus an alternation of two light/dark periods (12:12 L:D and 8:16 L:D), on the cell cycle of *T. lutea* cultures. For each photoperiod tested, an initial culture (cellular concentration around 2,000,000 cells mL^−1^) was used to inoculate three experimental cultures (biological replicates) at 250,000 cells mL^−1^ in a 150 mL volume continuously bubbled with 0.22 µm filtered-CO_2_-air mix in 250 mL sterile flasks. Cultures were maintained at regulated temperature (21 °C) and irradiance (140 µmol m^2^ s^1^). Experimental cultures were then left to grow and acclimate to each photoperiod for three days before analysis of the cell cycle. On the third day, 3 mL samples of each replicate and photoperiod tested were taken every hour for 8 h to study the cell cycle.

### 2.4. Chemical Blocking of the Cell Cycle

A second set of independent experiments was performed to block the cell cycle. To that aim, several chemical inhibitors were used as synchronizing agents to obtain *T. lutea* cultures in which most cells divide simultaneously (high cell count at G_2/M_ phase).

Three drugs were used: The DNA replication inhibitors hydroxyurea (HU, Sigma-Aldrich, St. Louis, MO, USA, reference 400046) and aphidicolin (APHI, Sigma-Aldrich, reference 178273) have been used to block cells after G_1/S_ transition, leading to cells being stopped at the S/G_2_ phase. In addition, the microtubules’ depolymerization was targeted by nocodazole (NOCO, Sigma-Aldrich, reference 487928) to obtain cells blocked at the M phase. These drugs were selected based on extensive literature demonstrating their effectiveness in synchronizing cell cycles in species related to *Tisochrysis lutea*, providing both proven efficacy and concentration guidance for our experiments. Preliminary studies helped determine optimal concentrations that induce cell cycle blocking without compromising cell viability, as confirmed by flow cytometry. Additionally, tests showed a synergistic effect between these agents and the light/dark cycle. Thus, all manipulations during the dark phase were performed under green light to avoid photosynthesis induction in our cultures. All experiments were conducted under the same conditions: the cultures were grown in 100 mL sterile Erlenmeyer at 21 °C, 140 µmol m^2^ s^−1^, and 8:16 L:D photoperiod. For NOCO blocking, NOCO was used on triplicate 50 mL cultures containing around 1 × 10^6^ cells mL^−1^, at final concentrations of 0 (controls: NEG-CTRL), and a range from 40 ng mL^−1^ to 10 µg mL^−1^. Incubations with NOCO were started 30–60 min before the beginning of the dark phase and lasted 8, 31, or 53 h, thus including up to twice the whole 8:16 L:D photoperiod. No removal of NOCO was performed. Samples of 4 mL of each replicate and condition were taken every two hours during the first 8 h of each dark phase to analyze the cell cycle of the cultures by flow cytometry and then once every 24 h.

For HU blocking, three conditions were tested in triplicate cultures: a drug-free negative control (NEG-CTRL), an exposure to HU (Cf = 0.64 µg mL^−1^) for 1 h (HU-1h-POS), and an exposure to HU (Cf = 0.64 µg mL^−1^) for 15 h (HU-15h-POS) before HU removal from the medium. The nine experimental cultures (Vf = 30 mL), which originated from the split of a 500 mL unique mother culture in the exponential growth phase continuously bubbled with 0.22 µm filtered-CO2-air mix (Cf = 2,000,000 cells mL^−1^), were maintained in 50 mL Erlenmeyer flasks. For the 15 h incubation, HU was added seven hours after the beginning of the dark phase (Day 0, late blocking), while for the 1 h incubation, HU was inoculated at the end of the light phase (Day 0, early blocking). Once the incubation was completed, after 1 h or 15 h for the early and late blockings, respectively, the drug was withdrawn according to the same procedure as for the APHI blockage. Then, the cultures were put again in the thermostatic chambers to go on with their cellular cycle. For flow cytometry analysis, 3 mL samples of each culture were collected during two consecutive days, specifically at the end of the light period (L7h), during the dark one at the peak of cell division (D6h), and in the middle of the dark phase (D9h). Concurrently, 0.1 mL samples were used to assess the cell density of the cultures.

For APHI blocking, three conditions were tested in triplicate cultures: a drug-free negative control (NEG-CTRL), a positive control with APHI (Cf = 1 µg mL^−1^) permanently in the medium (APHI-1_POS-CTRL), and a condition in which the drug was removed from the medium after 24 h of incubation (APHI-1_washed). The nine experimental cultures (Vf = 50 mL), which originated from the split of a 500 mL unique mother culture in the exponential growth phase (Cf = 1,000,000 cells mL^−1^), were maintained in 150 mL flasks continuously bubbled with 0.22 µm filtered-CO_2_-air mix. The drug was added to the cultures of both APHI-1_POS-CTRL and APHI-1_washed at the beginning of the dark phase on Day 0. After a 24 h incubation, cultures from condition APHI-1_washed were released from the chemical blockage by two washes via 20 min centrifugation at 900× *g* and 4 °C and resuspension in 50 mL fresh medium. Then, 5 mL samples were taken in each of the 9 experimental cultures at regular intervals throughout overall 23 h monitoring (Appendix A) to study the effect of the chemical blockage on the growth phase G1 (i.e., light exposure) and/or on the cell division (i.e., the dark phase).

### 2.5. Commitment Experiment

This part aimed to characterize the control features (“sizer” and “timer”) of the commitment point of the G_2/M_ phase transition. For that purpose, two staggered thermostatic chamber incubators were regulated at constant temperature (21 °C) and irradiance (140 µmol m^2^ s^1^) with an L:D regime of 8:16. The first incubator (A) was configured to have the light phase scheduled from 02:00 a.m. to 09:59 a.m., followed by the dark phase extending from 10:00 a.m. to 01:59 a.m. Meanwhile, the second incubator (B) was programmed to have the light phase from 10:01 a.m. to 06:00 p.m. and the dark phase from 06:01 p.m. to 10:00 a.m. Four days before the experiment, a unique 1500 mL mother culture was placed in the first incubator (B) to acclimatize to growing conditions. Afterward, this culture was divided into eighteen 75 mL experimental cultures (triplicates) in 100 mL Erlenmeyer flasks at a target concentration of 2,000,000 cells mL^−1^ and placed in incubator B for one-day acclimatization. For the commitment assay, cultures were submitted to six different photoperiods ranging from 0 h of light to the usual 8 h of light in a 24 h cycle (0 h; 2 h; 4 h; 6 h; 8 h). To achieve that, the experiment began exactly at the beginning of the dark in incubator A, which was also the beginning of the light in incubator B (10:01 a.m.). At 10:00, cultures from condition 0 h were moved from B to A to continue their dark phase. After the corresponding light durations in incubator B, cultures from conditions 2 h, 4 h, 6 h, and 8 h were moved in turn to incubator A to stop light energy input. Three mL samples were taken at regular intervals for 16 h to study cell cycle phase changes (Appendix A). For the sampling of cultures in the dark phase, green light was used to ensure that photosynthesis would not be triggered.

### 2.6. Cell Size

For the commitment experiment, cells were analyzed immediately after sampling: 100 µL of each of the eighteen experimental cultures were diluted in 9.9 mL of 0.2 µm filtered seawater; cell concentrations and cell diameter distributions were determined using a Multisizer 3 Coulter Counter (Beckman Coulter Inc., Brea, CA, USA). Because *T. lutea* cells are axially symmetrical, the estimated cell volume (Ve) was then calculated with the following formula:$$Ve=\left(\frac{3}{4}\right)\pi .\;{\left(\frac{dx}{2}\right)}^{3}$$
where *dx* is the cell’s diameter.

### 2.7. Critical Volume Calculation and Timer Estimation

The critical cell volume for division, termed the “sizer”, was estimated from the division probabilities observed. We estimate the percentage of division with a matrix population model based on that of Hunter-Cevera et al., 2013 [51]. These division probabilities, *δ*, were determined under the assumption that cells within each size class V_j_ divide into two during a discrete time interval Δ*t* and that the division probability can be quantified as the difference in cell counts E_x_ (of volume V_j_) between two successive time points.
$$\delta \left(t,Ve\right)=\left(\frac{Et-Et+1}{Et}\right)$$

Only measurements taken after the onset of the dark phase were included in this analysis. For each light exposure condition (2, 4, 6, 8 h), division probabilities were computed and then compared against a model fit using Hill’s equation to evaluate the relationship between division probability and critical volume.

The critical volume represents the minimum cell volume at which a positive division rate is observed, indicating the capacity of cells to initiate division. For the timer estimation, samplings were performed at regular times during the night, enabling visualization of the required time after the CP for the cells to divide (i.e., an increase of division rate for dividing cells).

### 2.8. Cell Cycle Analysis by Flow Cytometry

For each experiment followed by the cell cycle analysis, at each sampling time, 3 to 4 mL (depending on cell concentration) of culture samples were centrifuged at 900× *g* for 20 min at 4 °C. After supernatant removal, cells were fixed overnight using 10 mL of 70% ethanol. The day after, ethanol was removed from samples via a two-stage PBS washing where cells were centrifuged for 10 min at 900× *g* and 4 °C and resuspended in PBS (1st resuspension in 10 mL, 2nd resuspension in 200 µL). To prevent disturbance of DNA staining by the presence of RNA in the cells, an RNAse (RNase A, Sigma-Aldrich, reference 556746) (Cf = 10 µg mL^−1^) digestion step was performed on all samples at 37 °C for 30 min. Samples were then stained using Propidium Iodide (PI, Sigma-Aldrich, reference 537059) (Cf = 10 µg mL^−1^) or SYBR Green I (Cf = 1 µg mL^−1^) (depending on the experiment) during a 5 min incubation step at room temperature, in darkness, before analysis. The stained samples were analyzed using a MACSQuant Analyzer 10 flow cytometer (Miltenyi, Bergisch Gladbach, Germany) to obtain relative measurements (in arbitrary units) of the nuclear DNA content of the cells by the mean of their red (PI) or green (SYBR Green) fluorescence values after excitation by a blue laser (488 nm). Channel B3 (655–730 nm) was used for the PI staining, and channel B1 (BP 525/50 nm) was used for the SYBR Green I staining. Relative percentages of cells in the G_2/M_ (cell division phase) or G_0/1_ (growing phase) phase were determined after analysis of the cytograms using the FlowLogic v8.7 software (Inivai Technologies, Melbourne, Australia).

## 3. Results

### 3.1. Synchronization of T. lutea Cultures by L:D Cycles

The relative DNA content of the cells submitted to the three photoperiods was analyzed by flow cytometry to determine the proportion of cells in two phases of the cell cycle: G_0/1_ and G_2/M_. In this study, we hypothesize that synchronization at the culture level is indicated by significant fluctuations in the proportion of cells in the G_2/M_ phase between sampling points, capturing distinct shifts from division phases to periods of replication or growth. This approach aimed to determine whether photoperiod manipulations can synchronize cells efficiently, contrasting these findings with other methods.

Regardless of the photoperiod tested, *T. lutea* cells were predominantly in the G_0/1_ phase during the 7 h monitoring (Figure 1). In cultures exposed to continuous light, the G_2/M_ phase cell rate was maintained at 12.00% ± 1.10, with minimal fluctuations, indicating limited synchronization in division events (Figure 1). When the light period was shortened and alternated with a dark period of the same duration (L/D 12:12 h) (Figure 1), a slight increase in synchronization was observed, as the proportion of cells in the G_2/M_ phase slightly increased from 13.57% ± 0.77 near the end of the day (hour 0) to 20.74% ± 0.91 during the night (hour 6).

This synchronization effect became more pronounced under an L:D cycle of 8:16, where the culture was subjected to shorter day phases. During the day, cells in the G_2/M_ phase constituted less than 10% of the culture (hour 1). However, this proportion steadily increased throughout the night and appeared to reach a plateau, doubling by the sixth hour (19.36% ± 0.17).

### 3.2. Synchronization of T. lutea Cultures by Blocking Agents

In an attempt to increase the level of synchronization observed under L:D alternations alone, drugs known to block cells at different stages of the division process were tested on 8:16 L:D acclimated cultures. The use of alternating L:D cycles is based on prior experiments (not presented here) that demonstrated a synergistic effect between the photoperiod and drug treatment, leading to an increased accumulation of cells in the G_2/M_ phase at the maximum rate observed.

To assess each drug’s efficacy, we measured the proportion of cells that accumulated at the targeted cell cycle phase upon treatment, contrasting these findings with untreated controls. This outcome suggests that NOCO was ineffective for synchronization in *T. lutea* under the conditions tested. The first one, NOCO, induced no change in the DNA content distribution compared to the control, regardless of incubation time or exposure concentration (Figure 2). When cells were treated with HU, the majority of the culture was in the G_0/1_ growth phase. In comparison to control cultures (NEG-CTRL), cells treated with HU exhibited a higher percentage of cells in G_2/M_ recorded overnight (26.24% ± 1.76 vs. 15.76% ± 0.45 for NEG-CTRL, Figure 2). This increase suggests that HU treatment partially enhanced synchronization during the dark phase, aligning with the hypothesized synergistic effect between drug treatment and photoperiod alternation.

Surprisingly, after the release of APHI, over 90% of *T. lutea* cells were arrested in the G_2/M_ phase following 24 h of incubation (Figure 2). As expected, the proportion of cells in the G_0/1_ phase remained around 10%, in contrast to the concomitant increase in the G_2/M_ phase, confirming a highly synchronized culture (Appendix A). However, this high synchronization did not persist post-treatment. Once APHI was washed out (Figure 3), a progressive decrease in the G_2/M_ cell rate from 91% in the first third of the night (5 h after release) to a minimum of 41% at the end of the experiment (23 h post-release) was observed (Figure 3).

### 3.3. Determination of Commitment Point

To determine the light dose necessary for cells to reach the commitment point, cultures were exposed to different light durations, and monitoring of cell volume was conducted over the exposures.

Sixteen hours of dark acclimatization were conducted before the beginning of the experiment to ensure that all the cells were in the same physiological state.

Across all replicates and light exposure conditions, the probability of cell division increased steadily with cell size for cells larger than 50 µm^3^, reaching a plateau at 75 µm^3^, where division probability remained constant (saturation) (Figure 4). Additionally, within the size range of 50–75 µm^3^, the probability of cell division was influenced by the duration of light exposure. The longer the light exposure prior to darkness, the maximum probability of division. Indeed, the longer the prior light exposure, the higher the maximum probability of cell division, reaching up to 60% after 8 h of light. When light exposure was reduced to 6, 4, and 2 h, the maximum division probabilities decreased to 40%, 35%, and 25%, respectively (Figure 5). Furthermore, the probability of cell division was not constant throughout the night. In all conditions, the division probabilities during the first 4 h of darkness (0–4 h, Figure 4) were lower than those in the later hours (4–8 h and 8–12 h, Figure 4).

## 4. Discussion

The primary objective of this article is to provide a pioneering description of the *T. lutea* cell cycle. Simultaneously, we aim to devise a methodology for inducing synchronization within cultures of this microalga using either light exposure or chemical inhibitors.

### 4.1. Natural Synchronization

For phototrophic species, energy is acquired through photosynthetic electron transport, and the absence of photosynthesis in darkness is supposed to synchronize the cell division [38]. Synchronized cultures are crucial in cell cycle studies, ensuring a controlled setting for precise analysis and reliable interpretation of cell cycle events. Previous studies on diatoms [52] and unicellular green algae [30,53,54] have shown that alternating light and darkness can keep cultures highly synchronized, with 95–100% of cells dividing indefinitely, even without external stimuli.

The light/dark cycles employed in this study were derived from optimal growth conditions for *T. lutea* as used in biotechnology [55] (i.e., continuous light), as well as reflecting the light fluctuations in natural environments (i.e., L/D 12:12 h, 8:16 h). This approach allows simultaneously for an evaluation of the cell cycle behavior and culture synchrony in both controlled and realistic scenarios from the environmental point of view. Extending the duration of darkness from 0 h to 12 h and 16 h resulted in a higher increase in the cell proportion undergoing simultaneous division in the G_2/M_ phase. It is noteworthy that when the duration of light exposure is shorter (i.e., 8:16 h), the cell cycle is restricted in time. As a result, these cultures exposed that the most significant increase in the number of G_2/M_ cells occurs between the light and night phases. However, this subset still constituted a minority within the overall culture, comprising a maximum of 20%. The low rates of cells in the G_2/M_ phase in the present study are similar to findings by Farinas et al. (2006) [41] in the Chlorophyte *Ostreococcus tauri:* extending the photoperiod to 14 h made it possible to enhance the cell proportion of *O. tauri* cultures in the G_2/M_ phase, but it was limited to 20% of the whole culture.

Cell division is thus assumed to occur asynchronously, possibly manifesting in wave-like patterns. Such a pattern was found in *O. tauri* cultures [41]: when submitted to a 12:12 L/D cycle, cells were shown to divide once each during the experiment but not simultaneously in the whole culture. Two waves of division were evidenced by the authors, the first one at the end of the light phase and the second one at the beginning of the dark period. The explanation may lie in the notion of “group synchrony” described by Senger (1961) [56], who attributes this asynchrony to the presence of distinct groups of cells exhibiting varying numbers of divisions (i.e., number of daughter cells liberated from individual mother cells) within the same culture. According to this, it seems not uncommon to have, in a culture, the coexistence of groups of cells not dividing at the same moment. For instance, Soeder and Ried (1962) [57] highlighted the persistence of different cell groups in a “synchronized” culture in the green microalga *Chlorella* sp., even with L/D alternation treatment. Consequently, the individual cell cycles give rise to oscillations at the population level [58]. This phenomenon is noticeable in our study, where each *T. lutea* cell may demonstrate more or less autonomous behavior. Inducing synchronization indirectly through physical parameters such as light is supposed to provide a straightforward, low-tech, and, most importantly, reproducible method for achieving highly synchronized cultures. However, as shown in the present study and others [41,59], the effectiveness of this approach varies depending on the biological model used and may be inconsistent.

Another factor that may help explain this variability is highlighted in the work of [60], who suggested that polymorphisms in *T. lutea* drive cell-specific responses, notably impacting lipid accumulation and resilience under controlled light and nutrient conditions. This genetic diversity likely enhances each cell’s adaptive flexibility to the immediate microenvironment, which can lead to asynchrony in growth cycles. Such inherent variability complicates efforts to achieve full synchronization across cultures under standardized conditions.

Another aspect open for discussion is the overall duration of our light-dark (L/D) cycle, given the presumed 24 h cell cycle length of *T. lutea*. However, ample evidence suggests that the average duration of the cell cycle varies significantly depending on factors such as species, growth conditions, and environmental factors. For instance, in another haptophyte species, *Emiliania huxleyi*, cell division occurs within a 6 h window out of 24 h, influenced by variations in light and temperature conditions [61]. Based on our findings, it appears that synchronization is heightened by reducing the photoperiod (as compared to continuous light), while the duration of subsequent darkness is equally crucial for regulating cell division timing. It is evident that not all cells progress uniformly through the various stages of the cell cycle, particularly in the G_2/M_ phase. Another way to indirectly induce synchronization lies in the use of temperature [62,63]. In such cases, a temperature is chosen to block cell growth, and a subsequent release is induced by a return to a physiological temperature. This interesting trail has not yet been explored in *T. lutea*. To avoid the drawbacks of manipulating physical parameters and to improve the chances of better synchronization, an alternative is the use of commonly employed chemical methods for inducing synchrony in plants.

### 4.2. Chemical Synchronization

It is worth noting that cell proportions in the G_2/M_ phase in chemically untreated control groups can exhibit variability across experiments, even when culture conditions remain consistent aside from the tested chemical treatments. However, the consistency observed across our triplicate cultures assures us of the reliability of the data obtained.

Employing chemical inhibitors of the cell cycle triggers reversible cell cycle arrest, streamlining culture, and the synchronization of cultures without the necessity for expensive equipment. It makes it possible to produce synchronized cells essential for comprehensive studies across diverse biological models. It was demonstrated that molecules like NOCO, APHI, and HU were useful drugs for obtaining rapidly highly synchronized Chinese hamster ovary (CHO) cell populations in mitosis [64]. NOCO inhibits mitosis predominantly by affecting the dynamics of microtubule polymerization [65,66]. Although NOCO was administered at concentrations akin to those applied to the diatoms *Phaeodactylum tricornutum* (2.5 mg L^−1^ [67]) and *Cylindrotheca fusiformis* (1 nmol.mL^−1^ [59]), its application in the present study did not lead to a rise in the proportion of *T. lutea* cells arrested at the G_2/M_ stage. Yet, NOCO’s effectiveness as a synchronizing agent varies markedly across biological models: for instance, human cells [68,69] and *Saccharomyces cerevisiae* yeast [70] achieve synchronization rates surpassing 95%, whereas the previously cited diatoms (*C. fusiformis* and *P. tricornutum*) typically exhibit maximum synchrony between 23–50% [59,67].

Since blocking the majority of the culture before the M division phase proved ineffective, we explored a novel cellular target: DNA replication (S phase). HU and APHI are known as DNA synthesis inhibitors in plant cells [71,72]. HU inhibits the activity of ribonucleoside diphosphate reductase, thus depriving the cells of newly synthesized deoxyribonucleoside triphosphates, consequently preventing the new DNA strand construction [73]. In this study, we observed cell cycle arrest in HU-treated cells with a higher proportion of cells in the G_2/M_ phase (HU-15h-POS: 26%) compared to untreated cells (CTRL-NEG: 15%). However, they still constituted a minority of the total cell population. Similar partial cell inhibition was noted in the multiple fission dividing microalga *Chlamydomonas* spp.: HU at 5 nM inhibited solely the second mitosis while leaving the first unaffected [74]. Planchais et al. (2000) [40] attributed the diminished efficacy of HU on plant cells to their unique cell wall, which functions as a physical barrier, impeding the entry of the chemical agent. Electron microscopy observations conducted by Bendif et al. (2013) [3] unveiled the presence of organic scales enveloping the membrane of *T. lutea* cells, which can decrease their permeability to a variety of molecules.

We redirected our focus to another DNA replication inhibitor, known for its rapid reversibility: APHI [40,75]. APHI enables a large number of cells to enter the M phase from an S arrest induced by blocking DNA polymerases ***α*** and δ [40,76,77]. According to our results, APHI appeared to synchronize cellular divisions of *T. lutea* cells at 1 µg mL^−1^ concentration (>90% cells in G_2/M_ phase;). However, synchronization within the division phase steadily declined following the initiation of APHI washing, persisting at only 50% of the culture by the conclusion of a cell cycle (24 h), instead of transitioning synchronously to the subsequent growth phase. This phenomenon was similarly documented by Caillard and Mazzolini (1997) [78] in *Arabidopsis thaliana* cells. These cells, deprived of phosphate and treated with aphidicolin, exhibited synchronization but only during one phase of the cell cycle (G_2_) after S phase blockage. This may result from slow and heterogeneous release signal transduction among cells, leading to population-level desynchronization [53]. Excessive drug concentration may have caused irreversible cellular damage, inhibiting subsequent growth and division [79]. It underscores a limitation of chemical synchronization. In addition, the drug may disrupt cells, causing frequent cytotoxicity [58] and unwanted mutations [31]. Consequently, cells do not progress uniformly through the cell cycle, thus failing to accurately reflect events in an ideally growing and undisturbed culture [58].

In summary, while both L:D cycles and chemical agents influence synchronization to varying degrees, APHI treatment in combination with L:D cycling demonstrated the highest initial synchronization (Appendix A). However, in our case, high synchrony (>90% of total cells) was hoped for in our cultures, in addition to maintenance lasting over several cell cycles: indeed, we were looking for the possibility of performing more precise studies of the M division phase, the stage of the eventual meiotic processes. This central process of sexual reproduction is therefore essential for species improvement through genetic crossing. When striving to attain synchronous cultures of *T. lutea*, we discovered that chemical synchronization methods, like natural synchronization, failed to achieve our goal. Only light/dark alternation yielded reliability, even though it reduced synchronicity levels across successive cell cycles. This underscores the intricate nature of light-dependent cell division regulation in *T. lutea*, which responds diversely to photoperiod variations. This is why we investigated the regulatory mechanism at the start of the division, especially at the commitment point (CP) between the phases G_1_ and S.

### 4.3. Commitment Point

After light exposure, *T. lutea* cells showed a significant increase in biovolume. In autotrophs, photosynthesis produces glucose, providing energy that supports cell growth and expansion [80,81]. In darkness, cells undergo physiological processes like respiration and division into daughter cells, leading to a reduction in cell volume [82]. The commitment point in the cell cycle ensures that cells reach a critical size and acquire sufficient nutrients before initiating DNA synthesis and division [15,81].

The minimum size required for division is achieved through adequate photosynthetic growth during the G1 phase and confirmed by a “sizer” mechanism [22,23,83]. Our results with *T. lutea* showed a strong correlation between biovolume and division rate, influenced by light exposure duration (Figure 4). Larger cells (>75 µm^3^) had a higher division rate compared to smaller cells, supporting the requirement for a critical size before mitosis. However, incomplete synchronization of culture division indicates variability in cell cycle timing, preventing full division across the population. Smaller cells (<50 µm^3^) were likely in early growth phases, not ready for the commitment point [84].

Previous studies on *Cyanidioschyzon merolae*, *C. caldarium*, and *Galdieria sulphuraria* identified a critical cell size that is two to seven times greater than daughter cells, due to multiple fission allowing up to 16 new cells from a single parent [15,85]. Unlike those organisms, *T. lutea* likely undergoes a single division, yielding two daughter cells. This study’s methodology relied on the volume difference to distinguish mother cells (MCs) from daughter cells (DCs), unlike the more precise methods used in other studies [44,82]. Despite challenges in differentiation, our approach provided consistent estimates of the sizer across experimental replicates.

In *T. lutea*, the timing of reaching the commitment point is also affected by light exposure duration, as shown by Farinas et al. (2006) [41] in *Ostreococcus tauri*. In our study, short light exposure (2 h) did not trigger widespread division. The maximum division rate during the dark phase was only 30%, indicating that limited light exposure restricted photosynthetic activity and subsequent cell division, aligning with findings in *Chlamydomonas reinhardtii* [34].

Longer light exposure led to active growth and mitosis during the dark phase, suggesting a critical light duration of approximately 4 h for initiating division. Despite light limitations (2–3 h), a small fraction of cells initiated division, implying that some cells still reached the critical size. However, they did not achieve the division rate seen in cells exposed to longer light periods, indicating that the sizer is not the only factor at the commitment point. Instead, a light-sensitive “timer” also regulates the timing of division, with insufficient light preventing cells from dividing [34,86].

The division timing in *T. lutea* depended on the duration of darkness. A dark period of 4 to 8 h after nightfall promoted efficient division, likely due to better synchronization of internal timers, while shorter periods (0–4 h) led to reduced and delayed division. These findings are consistent with Shoshani and Bernstein (1969) [87], who found a 4 h light-dependent predivision period in *Chlamydomonas moewusii*. This suggests that extended darkness enhances synchronized entry into the division phase, emphasizing the interaction between light/dark cycles and internal cell cycle regulation.

It is important to note that our *T. lutea* cultures were not fully synchronized, so the thresholds calculated are estimates. Under the longest illumination (8 h per cycle), growth observed during the night did not indicate full population doubling, implying that not all cells were divided. Consequently, our estimates of the commitment point are based on a subset of the population. Despite the limited information available on microalgae cell cycles, these results provide an initial insight into the mechanisms regulating cell growth and division in *T. lutea*. Although the precise molecular pathways governing cell division and critical size thresholds in *T. lutea* remain unclear, genetic modification methods—such as CRISPR-Cas9—could serve as powerful tools to elucidate the role of key cell cycle regulators like cyclins and cyclin-dependent kinases (CDKs). By refining our understanding of these regulators, such approaches could ultimately enhance the synchronization of microalgae cultures, optimizing growth and productivity.

## 5. Conclusions

The present study reveals the complexity of controlling the cell cycle of the haptophyte microalga *T. lutea* and understanding its machinery in relationship with light. Despite our efforts using natural or chemical synchronization, we did not obtain more than 20% of total cells synchronized in the mitotic phase. We determined the capability of *T. lutea* to generate new daughter cells at a specific juncture, termed the commitment point (CP). We provide, for the first time, an initial estimate of the commitment point characteristics in *T. lutea*, which include a minimum duration of light exposure (timer) of 4–8 h associated with a minimum average volume (sizer) of 50 µm^3^. Understanding the data concerning critical minimum size allows for a deeper comprehension of cell division dynamics and the subsequent mechanisms driving cell differentiation. This critical size plays a pivotal role in transitioning from a mitotic cycle to meiosis, as exemplified by the process of sexual reproduction in diatoms. It necessitates the restoration of size post-achievement of a critical minimum cell size [88]. Synchronized cultivation combined with knowledge of mechanisms governing division cycles aids targeted intervention at specific cell cycle stages and manipulating reproductive phases. This offers prospects in biotechnology issues, especially for genetic cross-breeding or genetic engineering of strains. Such knowledge empowers strategic cultivation and genetic improvement, fostering efficient harnessing of microalgae’s diverse applications in biotechnology.

## Figures and Tables

**Figure 1 cells-13-01925-f001:**
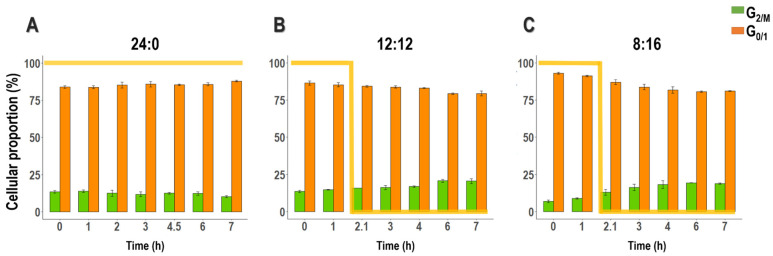
Evolution of the proportions (%) of *T. lutea* cells in each of the two G_0/1_ and G_2/M_ phases of the cell cycle, identified by flow cytometry, under different light: dark regimes (L:D): 24:0 (**A**), 12:12 (**B**) and 8:16 (**C**); (mean ± standard error, N = 3) throughout the experiment. Light level is represented by the yellow line; value 100 stands for daylight (140 µmol photon m^−2^ s^−1^), while 0 means complete darkness. Under continuous light condition (**A**), there is no noticeable variation over time in the proportion of cells in the G_2/M_ phase (green). However, with light-dark cycling (**B**,**C**), differences between the light and dark phases become more distinct, with the most pronounced peak occurring under the 8:16 cycle (**C**).

**Figure 2 cells-13-01925-f002:**
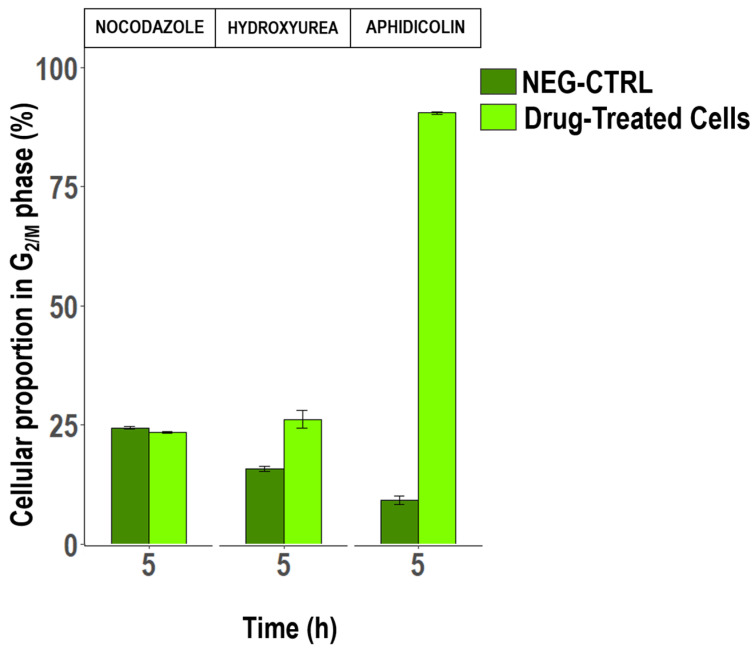
Variation in cell proportions of *T. lutea* in the G_2/M_ phase of the cell cycle identified by flow cytometry, after treatments with nocodazole (Cf = 10 µg.mL^−1^ after 53 h incubation), hydroxyurea (Cf = 0.64 µg mL^−1^ after 15 h incubation), or aphidicolin (Cf = 1 µg mL^−1^ after 24 h incubation), (mean ± SE, N = 3). Only the results obtained at the highest concentration and incubation time are shown, five hours after the beginning of the last dark phase. Each graph is accompanied by a control without exposure to the blocking agent (NEG-CTRL). The monitoring of cell proportion was carried out after the removal of the drug from the medium, with samples taken after x time (in hours; see Supplementary Data for details), except for NOCO where cells were kept in contact during the experiment.

**Figure 3 cells-13-01925-f003:**
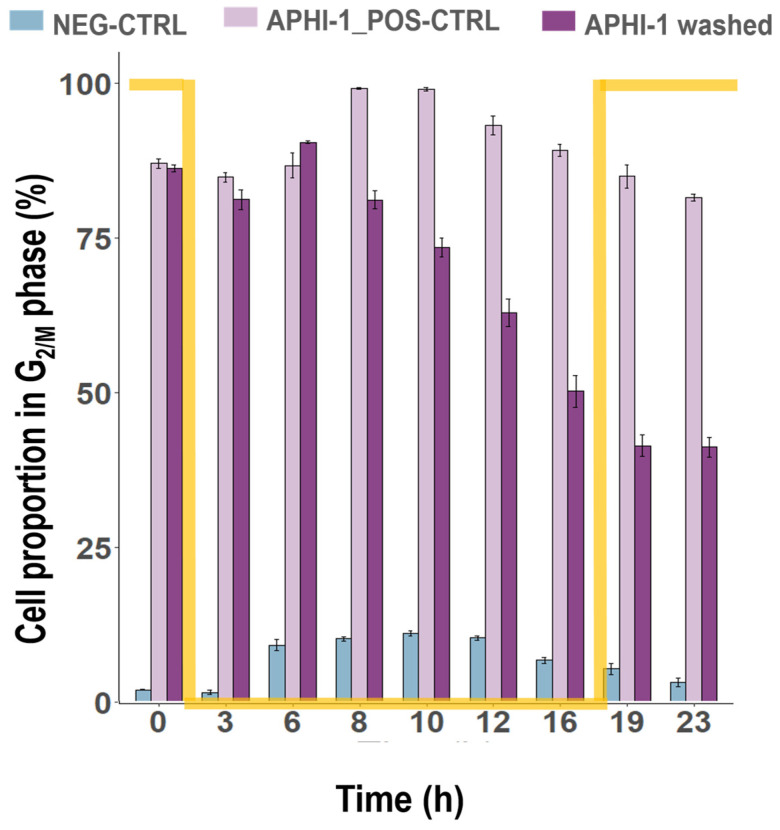
Variation in cell proportion of *T. lutea* cells in the G_2/M_ phase of the cell cycle identified by flow cytometry after different treatments with aphidicolin: APHI-1 washed: 1 µg mL^−1^; APHI-1_POS-CTRL: 1 µg mL^−1^; or in absence of treatment: NEG-CTRL (top). Twenty-four hours after the beginning of the experiment, when cultures were incubated for 24 h (APHI-1_POS-CTRL) or washed (APHI-1 washed), sampling was done for a further 23 h. (Mean ± SE, N = 3). The yellow line represents light level; values 100 and 0 correspond to daylight (140 µmol photon m^−2^ s^−1^) and darkness, respectively.

**Figure 4 cells-13-01925-f004:**
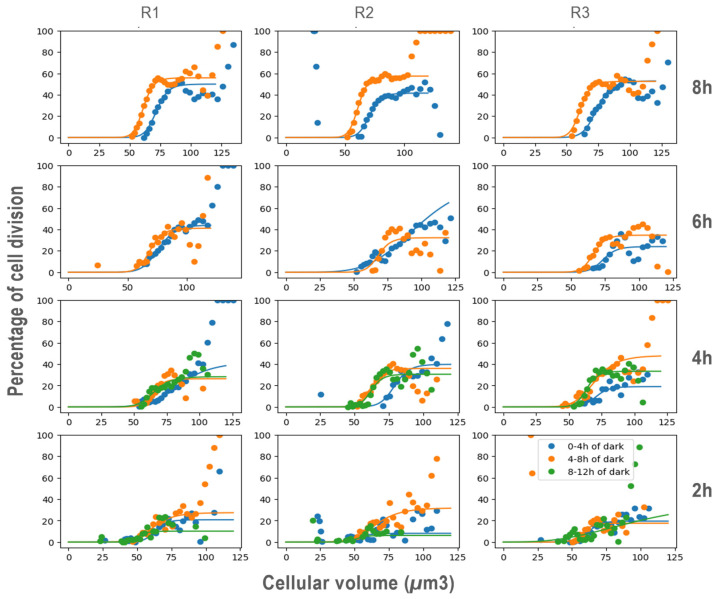
Evolution of cell division percentage as a function of cellular volume in *T. lutea* cells during the dark phase under varying previous light exposure durations. Each row in the figure corresponds to a specific light exposure duration: 8 h (top row), 6 h, 4 h, and 2 h (bottom row). The columns represent biological replicates (R1, R2, R3). Blue, yellow, and green dots denote the percentage of cells dividing at 0–4, 4–8, and 8–12 h of darkness, respectively. Fitted curves illustrate the variation of cell division probability relative to cellular volume, showing how changes in light exposure influence cell size regulation and division dynamics during the dark phase.

**Figure 5 cells-13-01925-f005:**
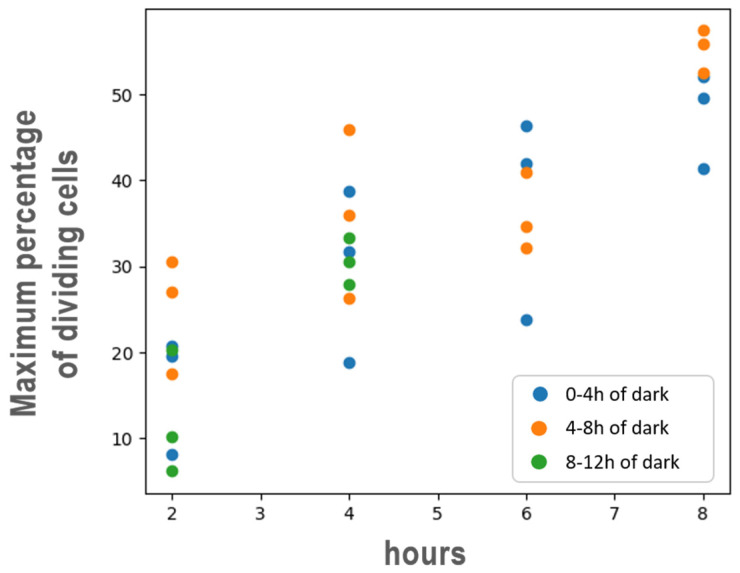
Impact of different light exposures (8 h, 6 h, 4 h, 2 h) on the maximum division probability of *T. lutea* cultures during the following dark period step (blue: 0–4 h, yellow: 4–8 h, green: 8–12 h).

## Data Availability

All raw data are available upon request by contacting the corresponding authors via email.

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
