# Peer review of "Cell Cycle Dynamics in the Microalga Tisochrysis lutea: Influence of Light Duration and Drugs"

_cells, 2024, doi:10.3390/cells13221925_

Round 1

Reviewer 1 Report

Comments and Suggestions for Authors

This MS studies the effects of natural and chemical synchronization methods to maximize their proportion at the division phase.  The MS covers interesting aspects that could contribute to the algae field, algae biotechnology and production which also the industry can benefit from.

The authors should address the following:

1.      The authors must discuss which transcription factors (TF) that are relevant to cell cycle are being changed/affected by the methods introduced in the MS.  It will strength the MS if the authors can show the relevant TF experimentally

2.      The authors should mention data regarding the S phase

Reviewer 2 Report

Comments and Suggestions for Authors

The manuscript presents valuable contributions to understanding the cell cycle of Tisochrysis lutea, with clear methodology and findings that are particularly relevant to biotechnological applications. To improve cohesiveness, presenting results as interrelated observations rather than isolated outcomes would strengthen the narrative. Based on these observations, major revisions are recommended, focusing on enhancing the interpretative depth in the results and discussion sections to draw more meaningful conclusions.

The abstract clearly outlines the study's focus on cell cycle regulation in Tisochrysis lutea. To strengthen the impact, consider briefly highlighting how findings on light duration and drug effects contribute to a broader understanding of cell cycle manipulation in this species. Including a concise statement on potential applications for controlled cultivation could make the abstract more engaging for readers.

The introduction provides a strong foundation on the ecological and biotechnological roles of microalgae, but the transition to the unique qualities of Tisochrysis lutea feels somewhat general. A more specific focus on the qualities that make Tisochrysis lutea an effective model for synchronization studies—such as its responsiveness to photoperiods—would provide a clearer rationale for this study. Additionally, explicitly framing how this work addresses gaps in existing research on microalgal cell cycles could reinforce its novelty.

The methodological section is detailed and well-structured, ensuring reproducibility. However, the choice of synchronization agents (Nocodazole, Hydroxyurea, and Aphidicolin) warrants further explanation, particularly considering the varying effects observed across different species. Providing background on how concentration levels were chosen or any preliminary testing conducted could enhance readers’ understanding of the experimental strategy. Additionally, since light manipulation is central to the study, briefly describing precautions taken against potential light variability or leakage would add robustness to this section.

The results section is thorough, but presenting findings in a way that builds a clear narrative would be advantageous. For example, contrasting the effectiveness of light/dark cycles with chemical treatments on synchronization could help highlight key patterns relevant to the study’s objectives. Reorganizing some findings to showcase how each synchronization method either aligns with or challenges the expected cell cycle progression would improve readability. A comparative summary figure could also be valuable here to visually capture differences across treatment groups.

The discussion addresses the findings thoughtfully. However, an in-depth analysis of why Tisochrysis lutea exhibited only partial synchronization under various conditions could reveal potential adaptive traits in fluctuating natural environments. Additionally, comparing this study’s findings with cell cycle dynamics in other algal species could be a valuable approach to understanding if T. lutea has unique regulatory mechanisms that resist synchronization. Speculating on future directions, such as the possibility of genetic modification to enhance synchronization, would align with the study’s applied objectives and provide readers with actionable research pathways.

How stable are the results across biological replicates in each treatment group?

Would including a summary of division percentages at each time point across the experimental photoperiod cycles enhance interpretability?

For cases where synchronization was achieved, is there a threshold duration in the photoperiod experiments at which the synchronization begins to decline?

Comments on the Quality of English Language

Proof read the manuscript to correct the grammatical errors.

Round 2

Reviewer 2 Report

Comments and Suggestions for Authors

Accepted.